# Construction and Validation of the Positive Mental Health Literacy Assessment Scale in Adults

**DOI:** 10.3390/ijerph20146391

**Published:** 2023-07-18

**Authors:** Cláudia Chaves, João Duarte, Francisco Sampaio, Joana Coelho, Amadeu Gonçalves, Vanda Santos, Carlos Sequeira

**Affiliations:** 1Superior School of Health, Polytechnic of Viseu, Rua Dom João Crisóstomo Gomes de Almeida, 102, 3500-843 Viseu, Portugal; cchaves@essv.ipv.pt (C.C.); jduarte@essv.ipv.pt (J.D.); agoncalves@essv.ipv.pt (A.G.); 2Nursing School of Porto, Rua Dr. António Bernardino de Almeida, 830, 844, 856, 4200-072 Porto, Portugal; franciscosampaio@esenf.pt (F.S.); carlossequeira@esenf.pt (C.S.); 3CINTESIS@RISE, Nursing School of Porto (ESEP), Rua Dr. Plácido da Costa, 4200-450 Porto, Portugal; 4Northern School of Health of the Portuguese Red Cross, Rua da Cruz Vermelha Cidacos-Apartado 1002, 3720-126 Oliveira de Azeméis, Portugal; 5Research Centre on Didactics and Technology in the Education of Trainers (CIDTFF), University of Aveiro, 3810-193 Aveiro, Portugal; vandasantos@ua.pt; 6Centre for Informatics and Systems of the University of Coimbra (CISUC), University of Coimbra, 3030-790 Coimbra, Portugal

**Keywords:** literacy, positive mental health, adults, community

## Abstract

Low mental health literacy levels have a profound negative effect on healthcare outcomes, usage of healthcare services and healthcare expenditure. To date, there is little research and a lack of instruments available to address and assess positive mental health literacy levels in community-dwelling adults. Thus, this study sought to develop an instrument to assess positive mental health literacy in adults and to evaluate its psychometric properties. A scale developed in European Portuguese and composed of 32 items was designed to assess positive mental health literacy among community-dwelling adults and was validated using exploratory factor analysis. Five latent factors were identified (decision making, prosocial attitudes, perception of resources, community involvement, and problem solving), whose hypothesised structure was confirmed by confirmatory factor analysis using structural equation modelling. Evidence showed that the scale’s factor structure is reliable and valid and adequately represents the theorised constructs. Thus, this may be a useful assessment tool for clinical practice since it will allow a more rigorous assessment of positive mental health literacy and better mental health promotion interventions in the population.

## 1. Introduction

Health literacy is a multidimensional social construct continuously evolving from a highly focused construct based on functional literacy (reading and writing) to one that focuses on the need for individuals to develop the knowledge, attitudes, and competencies necessary to become active agents of their health [1]. Thus, health literacy should be recognised to improve the health outcomes of the individual and populations [2]. Furthermore, mental health literacy is often regarded as a derivative of health literacy.

Promoting mental health and well-being in a specific community poses new challenges to healthcare workers. Poor mental health is considered a major national and global public health issue. Over the last decades, it has become a growing concern in Europe due to the significant overall prevailing mental disorders and the exponential increase in the consumption of psychotropic drugs [3]. According to the World Health Organization (WHO), the concept of health literacy has been used as a guide to mental health literacy, contributing to empowering the individual in decision making and self-care, resulting in significant health gains [4]. Literacy abilities are human resources that should not be underestimated, and literacy levels should be investigated and assessed in all linguistic populations [5].

Developing tools to assess mental health literacy levels evidencing the current understanding of the concept of mental health literacy is crucial to recognize, manage, or prevent mental health disorders and guide the development of mental health literacy interventions. Evidence shows that several mental health literacy assessment instruments have been applied in various countries, such as the United States of America, Australia, and the United Kingdom [1,6]. These instruments yield dimensions related to common knowledge. This knowledge includes the prevention of mental disorders, the ability to identify the first signs of mental illness, skills to implement self-help practices, the capacity to identify currently available treatments, self-help strategies to respond to mild or moderate problems, and ways of providing effective support to people with mental disorders [7]. Furthermore, this knowledge concerns actions meant to benefit our health and that of others.

The concept of positive mental health was first suggested by Marie Jahoda in 1958 [8], who developed a model with three domains: self-actualization—the ability of individuals to explore their potential; environment mastery; and autonomy, in the ability of individuals to identify, face and solve problems. In turn, these domains are divided into six general factors: attitudes towards oneself; growth and self-actualization; integration; autonomy; perception of reality; and interaction with the environment [9]. However, it was only in the 1990s that the concept started to receive closer attention. Thus, in 1999, Lluch Canut resumed the Multifactorial Model of Positive Mental Health, extending its applicability to clinical practice [10]. According to Lluch Canut’s model, positive mental health comprises six factors: personal satisfaction, prosocial attitude, self-control, autonomy, problem solving and self-actualization, and interpersonal relation skills [10,11]. These factors are influenced by individual traits like gender, race, ethnicity, and schooling and/or social/contextual factors, such as cultural background, resources in the community, and environmental events [11,12].

Identifying the opportunities and barriers to positive mental health literacy in community-dwelling persons is critical for adopting the necessary measures and bridging the gap between the identified problem and the specified goal [13]. Moreover, good family relationships, social support, social skills, healthy lifestyles, cultural values, religion, and productive interpersonal and group relationships contribute to positive mental health [14].

In this research, the Lluch Canut model, its factors, and its respective theoretical definition [11] underpinned the construction of the instrument to assess positive mental health literacy. An examination of the literature shows a knowledge gap in positive mental health literacy since the known instruments, such as the Mental Health Literacy Scale, the Mental Health Knowledge Questionnaire, or the Mental Health Knowledge Scale, only assess mental health literacy [1,12]. In an effort to combat this gap, the present study sought to design and assess the psychometric properties of a positive mental health literacy assessment instrument. It is important to note that this paper uses the definition of positive mental health literacy proposed by Bjørnsen et al. [7], which, in turn, uses Lluch Canut’s assumptions [15], i.e., the six factors of the Multifactor Model of Positive Mental Health mentioned above.

This study examined the research questions (a) what is the factor structure of the Positive Mental Health Literacy (MHL+) assessment tool obtained through exploratory factor analysis? and (b) is the factor model that is obtained through exploratory factor analysis confirmed through confirmatory factor analysis?

## 2. Methods

### 2.1. Design, Participants and Setting

Statistical procedures such as exploratory factor analysis (EFA) are often used in the evaluation and refinement of assessment tools. This set of techniques aims to find an underlying structure in a matrix and determine the number and nature of latent variables (factors) that best represent a set of observed variables [16]. According to Pasquali [17,18] and Marôco [19,20], factor analysis (FA) supports the assumption that empirical or observable variables (usually operationalised by indicators) can be explained by a smaller number of hypothetical variables commonly referred to as factors.

This psychometric study used a non-probability convenience sample with the following inclusion criteria: participants aged 18 or older who freely agreed to participate, residing in Portugal. There were no exclusion criteria. Data were collected from a large sample of at least seven subjects per item to avoid bias. The maximum margin of error for a 95% confidence interval was 4.89%. Data collection was carried out in primary health care, hospitals, schools, universities and universities for senior citizens, in Portugal, between May and December 2018 using paper-based questionnaires.

### 2.2. Instrument

Positive mental health is a complex construct encompassing several dimensions of mental health literacy. Those dimensions should be consistent with a broader construct of mental health literacy, including environmental, ecological, training and information, access to technologies, relational interactions with specific and extended groups, and help-seeking knowledge. In addition, the inclusion of societal components in the assessment dimensions of mental health literacy is deemed essential since research shows that societal changes, coupled with individual development, significantly impact help-seeking intentions for mental health and behavioural self-regulation.

These assumptions prompted the construction of a data collection instrument to assess positive mental health literacy. This Likert-type scale included 58 statements in the initial stage. Each statement included a five-answer option scale: (1) not at all important; (2) not very important; (3) somewhat important; (4) important; and (5) very important.

### 2.3. Procedures

#### 2.3.1. Procedure for Scale Construction

Specific items were developed to measure each construct effectively after identifying the scale’s constructs. The review of the scientific literature, namely the articles and instruments on positive mental health literacy, allowed the design of the items. The number of items included in the early version resonates with Pasquali [17,18], suggesting that the preliminary instrument should have three times as many items as the final instrument (what has already been produced on the topic is used as a reference); there are several identical items so that each factor has at least three items to support it. Finally, the inclusion of different expert panels and semantic analysis were considered to ensure content validity.

The first panel of experts consisted of ten experienced higher education research teachers in the mental health field and twelve specialised nurses in mental health and psychiatric nursing, each with more than five years of experience. They were asked to assess the relevance of the instrument’s items and whether there was a relationship between the item and the dimension being assessed. In addition, they were asked to provide additional information and suggestions to improve the items and ensure the psychometric quality of the scale. Each expert answered according to his/her opinion on the item and was able to suggest amendments to the items reviewed. After gathering the experts’ contributions, the descriptive analysis of the experts’ opinions was carried out to check the frequencies and percentages of agreement with the construct and the relevance of the item to the dimension assessed. The items rated ‘Good’ (above 75.0%) by all the experts were kept in the final instrument; the remaining items were either removed or corrected if considered very relevant to the construct’s assessment [21]. Following the requirements presented, the first panel of experts reduced the number of items on the scale from 204 to 134, meaning 70 items were eliminated at this stage. The 134 items comprising the positive mental health literacy assessment scale were reviewed by a second panel of four mental health experts to evaluate the relevance of each item within the instrument and determine whether there was a meaningful relationship between the item and the specific dimension it aimed to assess. This second panel of experts followed the same procedures as the first panel. After completing the procedure, 2 items were modified, and 76 items were excluded for lack of experts’ consensus on their validity or relevance.

Finally, a semantic analysis of the remaining items was performed to check for clarity of meaning. Therefore, a pre-test was conducted with 40 fourth-year undergraduate nursing students. This preliminary version was composed of 58 statements presented on a Likert-type scale, with each statement being rated according to a five-answer option, (1) not at all important; (2) not very important; (3) somewhat important; (4) important; and (5) very important. All modifications and suggestions presented were incorporated into the final version of the scale.

#### 2.3.2. Procedures to Perform Data Analyses

The metric properties of the Positive Mental Health Literacy Assessment Scale were then studied for validity and reliability. Validity studies included exploratory factor analysis and confirmatory factor analysis. The exploratory factor analysis was performed using IBM SPSS Statistics version 25, the principal component analysis, and Varimax orthogonal rotation. The Factor version 12.03.02 WIN 64 was also used as adjuvant software. Factor analysis of the polychoric correlation matrix was performed using Factor instead of Pearson correlation coefficient since the latter should only be calculated when the variables present asymmetry and kurtosis values with large biases and flattening. Concerning factor retention decisions, eigenvalues greater than 1 and the scree plot results were considered. For factor retention, the methods described in the IBM SPSS version and the parallel analysis using Factor were used. In addition, Promin oblique rotation was used. The Robust Diagonally Weighted Least Square (RDWLS) estimation was used with Factor. The IBM SPSS Amos version 25 was employed to conduct confirmatory factor analysis. The covariance matrix and the maximum likelihood estimation (MLE) algorithm for parameter estimation were used. This analysis considers two types of variables: manifest variables, also known as indicators, and latent variables, also known as factors or constructs, whose presence is indicated by their manifestation in the indicator or manifest variables [13]. Factor validity is a result of the quality of the overall fit of the factor model and the quality of local fit. The quality of overall fit was analysed according to the indices and respective reference values, namely chi-square (x^2^) degrees of freedom (x^2^/df) ratio, Comparative Fit Index (CFI) and Goodness of Fit Index (GFI), Root Mean Square Error of Approximation (RMSEA), Root mean square residual (RMR), and Standardized root mean square residual (SRMSR) [19]. According to Marôco (2014), the reference values for CFI, GFI, and TLI are a poor fit when <0.80, an acceptable fit when 0.8–0.9, a good fit when >0.9–0.95, and a very good fit when >0.95.

The quality of local fit was assessed using factorial loadings (λ) and individual item reliability. Factor saturation higher than 0.50 and individual reliability higher than 0.25 are suggested as reference values. The model fit was based on the modification indices (higher than 11; *p* < 0.001) produced by Amos and on different theoretical considerations.

Composite reliability (CR) measures the internal consistency of the items related to the factor. Values >0.70 suggest good internal consistency [22]. Convergent validity was determined using the average variance extracted (AVE) by checking the level of saturation of the items belonging to a particular factor. It is widely accepted that values higher than 0.50 suggest good convergent validity [19]. The discriminant validity of the factors was assessed by comparing the AVE of each factor with Pearson chi-square correlation. Evidence of discriminant validity is obtained when, for each factor, the squared correlation between the factors is lower than the AVE. Statistical analysis is implicit in the data analysis procedures, which include reliability and validity analysis. Other data analyses were performed, namely the determination of means and standard deviations. These measures and kurtosis values are inherent to the performed data analysis.

#### 2.3.3. Ethical Procedure

This study was approved by the National Commission for Data Protection (CNPD, Proc. No. 8837/2018). All participants were provided with the necessary information and signed informed consent. Their anonymity, data confidentiality, and autonomy were ensured. They were also informed that their participation was completely free, that they could withdraw from the study at any time, they would not receive any compensation, and that participating would not entail any harm or loss.

## 3. Results

A sample of 401 participants aged between 18 and 88 years (M = 43.0 years; SD = ±18.1) was recruited. Of the participants, 58.1% were female, approximately 96.4% were Portuguese, 97.6% were cohabiting, and more than half (53.9%) had completed at least upper secondary education. Moreover, 76.0% of the participants were employed, more than 61.0% had an average of one child, and 68.8% lived in a city.

The mean indices and corresponding standard deviations indicated that all items were well centred, as evidenced by scores higher than the reference value. Item 39, ‘Use of relaxation technique’, was the most problematic because of its high standard deviation.

Cronbach’s alpha per item was higher than 0.9, with an overall alpha of 0.958. Once internal consistency was determined, further validity studies and exploratory factor analyses of the scale were performed. In this first stage, factor rotation was not included. The Kaiser–Meyer–Olkin test (KMO = 0.932), which measures the adequacy of the sample size, suggested excellent sample adequacy. Combined with the anti-image correlation matrix, in which all variables had correlations above 0.5 (ranging from 0.864 to 0.969), and Bartlett’s test of sphericity, which is based on the chi-square distribution (x^2^ = 12,789.288; *p* < 0.01), the KMO test showed some relationships between the variables, suggesting further factor analysis. The Factor program revealed that the KMO test presented a very good result (=0.931), and Bartlett’s test of sphericity presented an x^2^ = 4373.0; *p* < 0.01. The MAS (Measure of Sampling Adequacy) values indicated that all items had values higher than 0.50 (reference value), suggesting they should be kept, i.e., they all measure the same construct.

The exploratory factor analysis was conducted using the principal component method, from which 12 factors emerged with eigenvalues greater than 1, explaining 63.52% of the total variance. The communalities were all higher than 0.40, ranging from 0.497 in item 8 to 0.814 in item 4.

However, the scree plot indicated the retention of five factors following the information retrieved from the slope of the curve. Hence, a new factor analysis with orthogonal varimax rotation was performed, forcing five factors (Figure 1).

The final penta-factor structure explains 50.8% of the variance after removing items 9, 11, 14, 18, 37, 41, 43, 46, 50, 31, 34, 36, 45, 15, and 35 because of communalities below 0.40 and after excluding items 8, 16 and 49 because of saturation values below 0.40. The Parallel Analysis, which is based on minimum rank factor and through the polychoric correlation matrix, showed five factors with eigenvalues greater than one and an explained variance of 56.468%. The goodness-of-fit indices in the exploratory factorial analysis were adequate (GFI = 0.984; AGFI = 0.991; RMSR = 0.0509).

The structure of the scale resulting from the EFA was not in line with the six-factor positive mental health model proposed by Lluch Canut. Thus, factors were named based on the semantic analysis performed with the items of each factor: factor 1—initiative and decision making, factor 2—self-concept and social interaction, factor 3—perception of resources, factor 4—community involvement, and factor 5—problem solving. The hypothesised penta-factor structure was submitted to confirmatory factor analysis, items’ trajectories to their corresponding factors and the respective estimates, critical ratios, and lambda coefficients. Evidence showed statistically significant critical ratios; therefore, all items could be kept.

The hypothesised initial model in Figure 2 shows the trajectory between the items and the factors. Factor loadings are high (λ ≥ 0.50), and individual item reliability is higher than 0.25. The overall goodness of fit indices is adequate for the chi-square degrees of freedom ratio (x2/df = 3.797); RMR = 0.037, RMSEA = 0.084 and SRMR = 0.072, and it is inadequate for GFI = 0.749 and CFI = 0.770. Since there is no consensus on using two subsamples—one for conducting exploratory factorial analysis and one for confirmatory factorial analysis—this study employed the factor retention technique based on resampling.

During the refinement of the initial model, specific errors were correlated based on the modification indexes suggested by Amos (Figure 3). After checking these correlations, adjustments were made to improve the overall model fit. Thus, six items were removed due to multicollinearity problems: items 58 and 51 (factor 1), items 42 and 27 (factor 2), item 20 (factor 3), and item 32 (factor 4). The overall goodness of fit indexes were adequate for the chi-square degrees of freedom ratio χ2/df = 2.388, RMSEA = 0.059, RMR = 0.033, SRMR = 0.062, and CFI = 0.901, and borderline acceptable for GFI = 0.860, confirming the model fit the empirical data.

Considering the high correlations between the factors and the theoretical proposal of a global score that could encompass the different scale dimensions, a second-order hierarchical construct was tested. This model included the positive mental health literacy factor (Figure 4).

The results suggest that the new model fit is good (χ2/df = 2.412; RMSEA = 0.059; RMR = 0.034; SRMR = 0.064; CFI = 0.900; GFI = 0.853). On the other hand, the factor loadings of the different dimensions of the second-order latent factor support the factorial validity. Therefore, the second-order hierarchical model can be accepted as the final model. The final result of the factor analysis using Factor corroborated the results obtained from IBM SPSS.

Table 1 summarizes the model fit indices of all the designed models, showing that the indices in the second-order model (final model) do not differ substantially from the model that included the modification indices and the removed items.

In this case, the fit is acceptable for GFI and good for CFI. TLI shows a poor fit of 0.890 for the refined model and the same value for the second-order model.

Regarding composite reliability, evidence shows the adequate consistency of all the scale dimensions with all indices above 0.70. As for AVE, the values observed do not indicate convergent validity as they show indices below 0.50, except for factor 3—Perception of resources (AVE = 0.546) (Table 2).

On the other hand, discriminant validity was only found in the correlation between factors (factor 1 vs. factor 4), (factor 2 vs. factor 4), (factor 3 vs. factor 4), and (factor 4 vs. factor 5).

The psychometric study of the scale was completed by measuring the internal consistency of the remaining items per subscale and the convergent/divergent validity of the different items.

The internal consistency analysis of each dimension showed good Cronbach’s alpha values for the ‘Initiative and decision making’ factor ranging between 0.843 in items 53 and 54, and 0.867 in item 30, with an overall alpha of 0.870. Item 55 had the highest correlation with ‘decision making’ (r = 0.732), but item 54 (58.5%) reached the highest variability.

The ‘Self-concept and social interaction’ dimension showed Cronbach’s alpha values between 0.755 for item 44 and 0.768 for item 28. The overall alpha value was 0.791. Item 40 scored the highest correlational value (r = 0.791), while item 44 had the highest variability of response (35.4%).

As for the internal consistency of the ‘Perception of resources’ subscale, alpha values ranged from 0.844 in item 24 to 0.875 in item 19, with an overall alpha of 0.877, suggesting good consistency. Item 24 had the highest correlation index (r = 0.756) and the highest variability of response (67.3%).

The analysis of factor 4, caring for oneself and getting involved in the community, showed internal consistency values with Cronbach’s alphas ranging between 0.800 (item 6) and 0.875 (item 5), with an overall alpha value of 0.837. In addition, item 6 showed the highest correlation index (r = 0.664), while item 39 had the highest variability of response (63.3%).

Finally, alpha values of factor 5, ‘Problem solving’, were considered reasonable, ranging from 0.739, in item 1, to 0.779 in item 12, with an overall alpha of 0.793. Item 1 showed the highest correlational value (r = 0.619), and item 4 had the highest variability of response (55.5%).

Pearson correlation was calculated between the different factors and the global value of the scale. The produced results indicated positive correlations between the five subscales and their global value, ranging between r = 0.307 (Problem-solving vs. Community involvement), explaining 9.4%, and r = 0.653 (Self-concept and social interaction vs. Initiative and decision making), with an unexplained variability of 57.3% (Table 3).

The correlations between the different subscales and the global value were higher and showed a variability above 48.0%.

Table 4 presents the convergent/discriminant validity of the items. All items in bold show positive and significant correlations with the respective scales (*p* < 0.01), indicating a good convergent/discriminant validity of the items. The global factor of the scale scored the second highest correlational value.

Finally, the Positive Mental Health Literacy Assessment Scale in adults was applied, and statistical analysis of each factor examined whether they were discriminated by some socio-demographic variables, such as gender and age. Since the subscales did not include the same number of items, they were ranked according to amplitude on a scale ranging from 0 to 100 to enable factor comparability. The formula was ((raw score-minimum expected value)/amplitude)) × 100. Table 4 summarizes the results of the statistical analysis. The lowest minimum score was for factor 4, and 100 was the highest score in all the subscales. The analysis of the average indexes showed the lowest average index in factor 4 (M = 66.4; SD = ±16.5) and the highest average index for factor 2 (M = 86.1; SD = ±10.5). The coefficients of variation showed low dispersion, except for factor 4, with moderate dispersion, and the skewness and kurtosis values revealed a right-skewed distribution and leptokurtic curves, except for factors 1 and 4, with normokurtic distribution.

This analysis aimed to determine whether gender discriminated against some positive mental health literacy subscales. The results showed higher literacy rates among female participants, statistical significance in factors 1, 2, and 3 and for the global factor, and no significance for factors 4 and 5 (Table 5).

On the other hand, concerning age, young people showed higher literacy scores for ‘Initiative and decision making’, ‘Self-concept and social interaction’, and ‘Problem solving’. In contrast, adults scored higher for the ‘Perception of resources’ and general literacy (Global factor), and older people scored higher on ‘Community involvement’. Factor 2 and the global factor were the only factors not discriminated by age (Table 6).

Finally, according to the 25th and 75th percentiles, cut-off groups were established for the global factor, and participants with low, moderate, or high positive mental health literacy were ranked accordingly. Since there is no consensus on using two subsamples (one for conducting exploratory factorial analysis and one for confirmatory factorial analysis), factor retention based on resampling was used in this study. According to the produced results, 26.9% of the participants had low mental health literacy, 44.1% had moderate mental health literacy, and the remaining 28.9% had high mental health literacy.

## 4. Discussion

This study provides an instrument to assess positive mental health literacy among the adult population (Appendix A). The only instrument developed to assess positive mental health literacy was validated in a sample of upper secondary school students [7], so the scale developed in this study is the first to allow assessing this construct in adult subjects. The final version of the instrument has 32 items distributed throughout five factors. The assessment tool presents a very good internal consistency, ranging from 0.791 in factor 2 (Self-concept and social interaction) and 0.877 in factor 3 (Perception of resources).

An extensive review of relevant literature guided the development of this instrument to enhance comprehension of the construct, definition, and associated dimensions. In addition, a group of experts was also recruited to critically analyse the possible elements/items to be included in the positive mental health literacy assessment instrument. Then, content validation was determined based on in-depth qualitative observation. Finally, the analysis of the instrument’s psychometric characteristics was performed with a non-probability convenience sample of 401 participants.

This work has allowed the design of a new data collection instrument. This instrument included 32 of the 58 items subjected to internal consistency and exploratory and confirmatory factor analyses. The remaining items were included in a second-order penta-factorial structure, which corroborates the available literature on mental health literacy [1,21], emphasising the need to develop tools that support multidimensional procedures to assess mental health literacy. Thus, factor 1 was called ‘initiative and decision-making’, factor 2 was termed ‘self-concept and social interaction’, factor 3 was named ‘perception of resources’, factor 4 was dubbed ‘community involvement’, and factor 5 ‘problem-solving’. All denominations were established by semantic analysis of the respective items. A global factor resulting from the sum of all items distributed by the factors was also generated.

The instrument structure was carefully planned to target this specific population using a brief instrument design with a simple and easy-to-understand language. The items’ content validity and semantic analysis by the panel of experts have largely contributed to enhancing the final results.

The available evidence on mental health literacy shows that the increase in the knowledge about mental health and mental disorders and the reduction of mental illness-related stigma at the individual, community and institutional levels are linked to a greater demand for health services [23]. In view of this reality, this study sought to design an instrument to be completed by respondents, regardless of their educational level, or that otherwise could provide the respondent with the opportunity to reach out for help from other people or institutions.

The sensitivity analysis of the scale items led to the exclusion of several items whose factor loadings were too close in two or more factors or too low in one of the factors. This items’ exclusion as a result of sensitivity analysis is in line with Daniel [24] (p. 136), whose study aimed at mapping the main techniques for measuring reliability and their algorithms, arguing that ’it might be advisable to delete items from a scale when it is being built or when one wishes to reduce the number of items. Otherwise, the deletion procedure strictly inhibits comparing our results with the original scale.’

Internal consistency analysis of the total item and inter-item correlations support the penta-factor model of this instrument. In fact, once the refinement of the instrument was completed, there was very good internal consistency (α = 0.898) for the scale’s global factor [25]. As for the different dimensions, internal consistency ranged between good and very good, with alpha values of 0.870 for factor 1 (decision making), 0.791 for factor 2 (prosocial attitudes), 0.877 for factor 3 (perception of resources), 0.837 for factor 4 (community involvement), and 0.793 for factor 5 (problem solving). However, further research should consider other validity and reliability tests, such as temporal stability (test-retest reliability) or concurrent validity. These analyses should use instruments that are already designed and validated for the Portuguese population to measure positive mental health literacy and that are expected to assess the constructs included in this instrument as a single instrument or coupled with others.

The exploratory factor model was validated by confirmatory factor analysis, revealing adequate goodness of fit indices after establishing the covariance between the errors of the items and the respective different factors. The establishment of this covariance is explained by the fact that the semantic contents of each item under study are close to each other and bear no relation to the possible existence of a small factor that was not considered.

On the other hand, all items revealed adequate factor loadings with coefficients higher than 0.50 [19]. However, although the composite reliability values were satisfactory or adequate for all factors, the low AVE values (all factors except factor 3, perception of resources, AVE = 0.546) suggested that the items’ behaviour is only partially explained by those given items.

This means that the factor structure should be carefully considered, as many dependent variables are commonly used in psychometric studies, especially in the design and validation of a new measurement instrument. For example, considering this study, the sample size must be examined in future studies. Also, further research should involve a broader and more heterogeneous population since the analysis of their diverse behaviours will likely increase the validity and reliability of the instrument. It should be noted that the systematic review of the literature by Wei [1] demonstrates that most studies are carried out with adult participants, and only four studies focus on adolescents. This highlights the need for developing, assessing, and validating tools that handle mental health knowledge and are specifically aimed at young people who are going through difficult times and may experience mental illness or disorders.

Regardless of the number of people interviewed, the total score of each construct was given the same weight for comparability purposes. Hence, and following the guidelines suggested by Daniel [24], during the validation of the positive mental health literacy assessment scale in adults living in the community, raw scores were coded and presented on a positive orientation (positivity) scale, ranging from 0 (lowest level of literacy) to 100 (highest level of literacy). These guidelines helped classify cut-off groups for the global score. Also, this allowed us to group participants into three levels according to their literacy scores: low, intermediate, and high.

The first potential limitation of this study is that the Positive Mental Health Literacy Assessment Scale is a self-report instrument that can bias responses. To minimize the appearance of the phenomenon of social desirability, anonymity was guaranteed when completing the instrument. The convenience sampling technique can also be a potential limitation, as it hinders the generalisability of the results. Finally, convergent validity was not assessed in this study; however, no instruments were found in the literature assessing the same construct, rendering it impossible to perform this assessment. Despite these limitations, we collected data from a large sample, which is one of the main methodological strengths of this study.

## 5. Conclusions

The final structure of the Positive Mental Health Literacy Scale included 32 items divided into five first-order factors and one second-order global factor.

The results showed that the instrument is promising and is a reliable and valid measurement tool to access positive mental health literacy in Portugal because of its content and factor structure. Therefore, this instrument can be recommended for studying Portuguese samples recruited from different contexts.

However, further research should be conducted to better explain the role of socio-demographic variables such as gender, age, and marital status, among others. Further studies will also have to confirm the invariance of the construct and whether it is suitable to be administered to adolescents. Finally, it would be interesting to assess the psychometric properties of the instrument in younger populations (children and adolescents).

## Figures and Tables

**Figure 1 ijerph-20-06391-f001:**
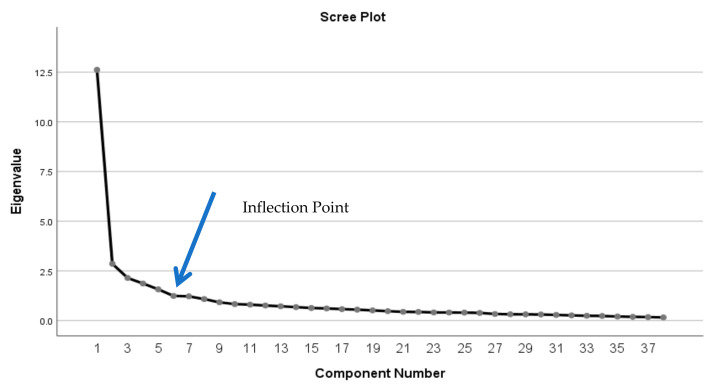
Scree plot.

**Figure 2 ijerph-20-06391-f002:**
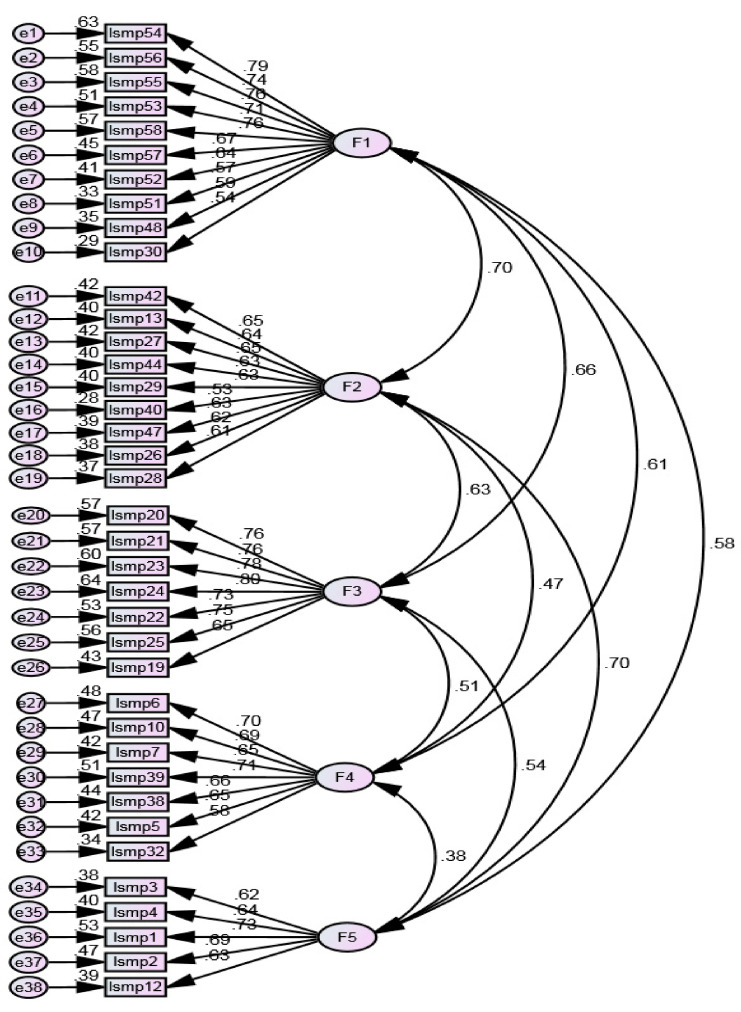
Hypothesised initial model.

**Figure 3 ijerph-20-06391-f003:**
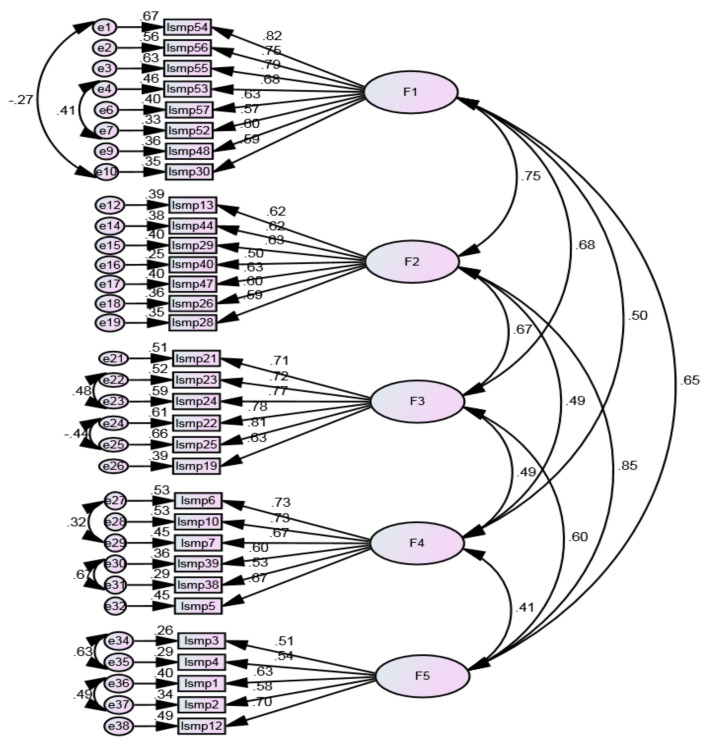
The model with the items removed and the modification indices.

**Figure 4 ijerph-20-06391-f004:**
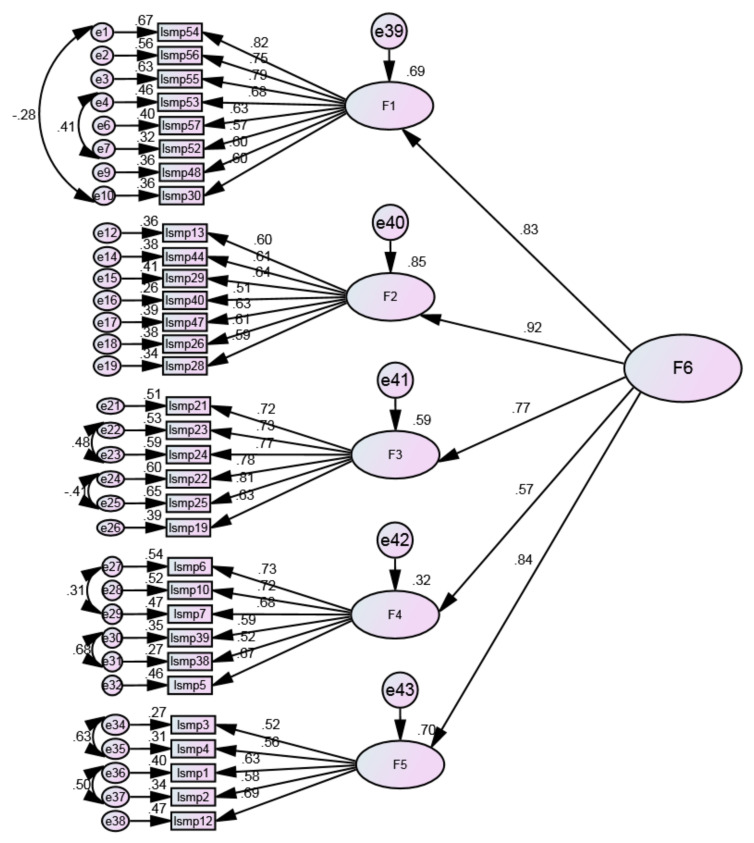
Second-order model (final model).

**Table 1 ijerph-20-06391-t001:** Model fit indices of the initial model, the model including the items removed and the modification indices, and the second-order model (final model).

Model	x^2^/df	GFI	CFI	RMSEA	RMR	SRMR
Initial model	3.797	0.749	0.770	0.084	0.037	0.072
Model including the items removed and the modification indices	2.388	0.860	0.901	0.059	0.033	0.062
Second-order model (final model)	2.412	0.853	0.900	0.059	0.034	0.064

x^2^/df—Chi-square (x^2^) degrees of freedom ratio. GFI—Goodness of Fit Index. CFI—Comparative Fit Index. RMSEA—Root Mean Square Error of Approximation. RMR—Root Mean Square Residual. SRMR—Standardized Root Mean Square Residual.

**Table 2 ijerph-20-06391-t002:** Composite reliability and average variance extracted.

Factors	CR	AVE
Factor 1 Initiative and decision making	0.874	0.469
Factor 2 Self-concept and social interaction	0.796	0.359
Factor 3 Perception of resources	0.877	0.546
Factor 4 Community involvement	0.820	0.435
Factor 5 Problem solving	0.732	0.356

CR—Composite Reliability. AVE—Average Variance Extracted.

**Table 3 ijerph-20-06391-t003:** Pearson’s correlation matrix between the subscales and the global factor.

Subscales	α	F1	F2	F3	F4	F5
Factor 1—Initiative and decision making	0.870	--				
Factor 2—Self-concept and social interaction	0.791	0.653 ***	--			
Factor 3—Perception of resources	0.877	0.597 ***	0.574 ***	--		
Factor 4—Community involvement	0.837	0.485 ***	0.396 ***	0.432 ***	--	
Factor 5—Problem solving	0.793	0.501 ***	0.591 ***	0.466 ***	0.307 ***	--
Global factor	0. 958	0.852 ***	0.811 ***	0.785 ***	0.719 ***	0.694 ***

*** *p* < 0.05.

**Table 4 ijerph-20-06391-t004:** Statistics concerning the positive mental health scale.

	Min	Max	M	S.D.	CV (%)	Sk/error	K/error
Factor 1—Initiative and decision making	34.38	100.00	79.23	11.99	15.13	−2.065	1.008
Factor 2—Self-concept and social interaction	39.29	100.00	86.16	10.57	12.26	−8.754	7.814
Factor 3—Perception of resources	25.00	100.00	83.23	12.73	15.29	−5.467	3.222
Factor 4—Community involvement	16.67	100.00	66.46	16.51	24.84	−2.786	−0.419
Factor 5—Problem solving	35.00	100.00	83.45	12.23	14.65	−6.860	4.748
Global factor	41.41	100.00	79.76	9.85	12.34	−3.516	2.662

**Table 5 ijerph-20-06391-t005:** UMW test between gender and positive mental health.

Gender	Female	Male	z	*p*
	Average Rank	Average Rank		
Factor 1—Initiative and decision making	220.90	173.40	−4.069	0.000
Factor 2—Self-concept and social interaction	216.94	178.89	−3.264	0.001
Factor 3—Perception of resources	219.38	175.51	−3.774	0.000
Factor 4—Community involvement	207.81	191.55	−1.391	0.164
Factor 5—Problem solving	206.10	193.92	−1.048	0.294
Global factor	219.26	175.68	−3.717	0.000

**Table 6 ijerph-20-06391-t006:** Kruskal–Wallis test between age and positive mental health.

Age	Young	Young Adult	Adult	Elderly Person	K-W H	*p*
	Average Rank	Average Rank	Average Rank	Average Rank		
Factor 1—Initiative and decision making	229.50	188.97	197.98	181.49	8.453	0.038
Factor 2—Self-concept and social interaction	211.54	193.92	209.56	170.64	5.905	0.116
Factor 3—Perception of resources	203.68	172.31	213.55	208.01	7.901	0.048
Factor 4—Community involvement	158.60	170.02	221.97	263.54	40.938	0.000
Factor 5—Problem solving	233.24	202.38	188.94	178.87	11.271	0.010
Global factor	203.96	178.31	210.76	205.60	4.807	0.186

## Data Availability

Not applicable.

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
