# Peer review of "Construction and Validation of the Positive Mental Health Literacy Assessment Scale in Adults"

_ijerph, 2023, doi:10.3390/ijerph20146391_

Round 1

Reviewer 1 Report (New Reviewer)

The manuscript focuses on the psychometric properties of the Positive Mental Health Literacy Assessment Scale, which I consider a very promising initiative. Researchers in positive psychology and related fields may be interested in the results that were found and reported.

I find the introduction to be well written and grounded, which is a good summary of the concept of mental health literacy.

My major criticism is the methodology for conducting the factor analysis of Positive mental health literacy scale:

1. Principal component analysis with varimax rotation is not the optimal approach. Using orthogonal rotation potentially results in a less useful solution where factors are correlated. The correlation between the factors is high: 0,31-0,65. “Perhaps the best way to decide between orthogonal and oblique rotation is to request oblique rotation [e.g., direct oblimin or promax from SPSS] with the desired number of factors (Brown, 2009) and look at the correlations among factors...if factor correlations are not driven by the data, the solution remains nearly orthogonal. Look at the factor correlation matrix for correlations around .32 and above. If correlations exceed 0.32, then there is 10% (or more) overlap in variance among factors, enough variance to warrant oblique rotation unless there are compelling reasons for orthogonal rotation” Tabachnick and Fiddell (2007). Please reconsider or justify your choice of rotation in the text.

2. It would be beneficial to check the number of potential factors using additional techniques in addition to the Kaiser criterion (eigenvalue >1.0) and scree plot: e.g., parallel analysis. Watkins (2018): „Although selection of the correct number of factors to retain is one of the most important decisions in EFA (Child, 2006; Fabrigar & Wegener, 2012; Gorsuch, 1983; Izquierdo et al., 2014; Norman & Streiner, 2014), the default method used by many statistical software programs (e.g., the “eigenvalue 1” rule) is usually wrong and should not be used (Fabrigar & Wegener, 2012; Izquierdo et al., 2014; Norris & Lecavalier, 2010). Measurement specialists have conducted simulation studies and concluded that parallel analysis and MAP are the most accurate empirical estimates of the number of factors to retain and that scree is a useful subjective adjunct to the empirical estimates (Velicer, Eaton, & Fava, 2000; Velicer & Fava, 1998).”

3. In the case of CFA, when considering possible structures, it would be important to illustrate and compare the fit of several solutions, e.g., unidimensional, correlated factors, second order, or bifactor models. The unidimensional model serves as a comparison. The bifactor structure includes the specific dimensions and the general factor, as used in the questionnaire evaluation (subscales and total score). We now see a correlated and second order models, although it would be more interesting to decide on the best model based on the fit of several models.

4. Since the questionnaire's overall score is also used, I believe the second-order or bifactor models are much more suitable. The bifactor structure contains several specific factors (e.g. 5) and one general factor. See for example: Reise, 2012.

5. The resulting fit indicators are poor, and it would be necessary to explain why correlations between errors exist. Modification indices often show that model fit would improve if one or more residuals among indicator variables were allowed to correlate. This practice is problematic for a variety of reasons. See for review: Hermida, R. (2015).

6. The fit index criteria you use are very lenient. More recently, though, much stricter indicators have been used. A CFI or GFI of 0.90 or below is not considered a good result. This will most likely cause problems when replicating the factor structure on another sample. „The results suggest that, for the ML method, a cutoff value close to .95 for TLI, BL89, CFI, RNI, and Gamma Hat; a cutoff value close to .90 for Mc; a cutoff value close to .08 for SRMR; and a cutoff value close to .06 for RMSEA are needed before we can conclude that there is a relatively good fit between the hypothesized model and the observed data. Furthermore, the 2-index presentation strategy is required to reject reasonable proportions of various types of true-population and misspecified models. Finally, using the proposed cutoff criteria, the ML-based TLI, Mc, and RMSEA tend to overreject true-population models at small sample size and thus are less preferable when sample size is small.”  Hu & Bentler, 1999.

In one of the more comprehensive and widely cited evaluations of cutoff criteria, the findings of simulation studies by Hu and Bentler (1999) suggest the following guidelines for acceptable model fit: (1) SRMR values close to .08 or below; (2) RMSEA values close to .06 or below; and (3) CFI and TLI values close to .95 or greater. Brown & Moore, 2012.

7. Additionally, it would be beneficial to evaluate and demonstrate the normality of the variables, as this may also influence the method of factor analysis chosen.

In my opinion, the above-mentioned additional analysis would greatly increase the value of this study.

References

Brown, T.A., Moore, M.T. (2012). Confirmatory factor analysis. Handbook of structural equation modeling. 361, 379

Hermida, R. (2015). The problem of allowing correlated errors in structural equation modeling: concerns and considerations. Computational Methods in Social Sciences, 3(1), 5-17.

Hu, L. T., & Bentler, P. M. (1999). Cutoff criteria for fit indexes in covariance structure analysis: Conventional criteria versus new alternatives. Structural equation modeling: a multidisciplinary journal, 6(1), 1-55.

Reise, S. P. (2012). The Rediscovery of Bifactor Measurement Models, Multivariate Behavioral Research, 47:5, 667-696.

Tabachnick, B. G., & Fidell, L. S. (2007). Using multivariate statistics (5th ed.). Boston, MA: Allyn & Bacon.

Watkins, M. W. (2018). Exploratory factor analysis: A guide to best practice. Journal of Black Psychology, 44(3), 219-246.

Author Response

Dear reviewer,

Thank you very much for your so important recommendations.

The manuscript focuses on the psychometric properties of the Positive Mental Health Literacy Assessment Scale, which I consider a very promising initiative. Researchers in positive psychology and related fields may be interested in the results that were found and reported.

I find the introduction to be well written and grounded, which is a good summary of the concept of mental health literacy.

Thank you very much for your gentle comment.

Principal component analysis with varimax rotation is not the optimal approach. Using orthogonal rotation potentially results in a less useful solution where factors are correlated. The correlation between the factors is high: 0,31-0,65. “Perhaps the best way to decide between orthogonal and oblique rotation is to request oblique rotation [e.g., direct oblimin or promax from SPSS] with the desired number of factors (Brown, 2009) and look at the correlations among factors...if factor correlations are not driven by the data, the solution remains nearly orthogonal. Look at the factor correlation matrix for correlations around .32 and above. If correlations exceed 0.32, then there is 10% (or more) overlap in variance among factors, enough variance to warrant oblique rotation unless there are compelling reasons for orthogonal rotation” Tabachnick and Fiddell (2007). Please reconsider or justify your choice of rotation in the text. 

Thank you very much for your recommendation. We have also used the Promin oblique rotation in the Factor software. The results corroborated (are in line) with the ones we obtained by using the principal component analysis with varimax rotation. That information was included in the Methods section (2.3.2) and in the Results section.

It would be beneficial to check the number of potential factors using additional techniques in addition to the Kaiser criterion (eigenvalue >1.0) and scree plot: e.g., parallel analysis. Watkins (2018): „Although selection of the correct number of factors to retain is one of the most important decisions in EFA (Child, 2006; Fabrigar & Wegener, 2012; Gorsuch, 1983; Izquierdo et al., 2014; Norman & Streiner, 2014), the default method used by many statistical software programs (e.g., the “eigenvalue 1” rule) is usually wrong and should not be used (Fabrigar & Wegener, 2012; Izquierdo et al., 2014; Norris & Lecavalier, 2010). Measurement specialists have conducted simulation studies and concluded that parallel analysis and MAP are the most accurate empirical estimates of the number of factors to retain and that scree is a useful subjective adjunct to the empirical estimates (Velicer, Eaton, & Fava, 2000; Velicer & Fava, 1998).”

Thank you very much for your important recommendation. We have now carried out parallel analysis using the Factor software, and the results regarding factor retention are in line with those obtained by using the Kaiser criterion and scree plot. That information was included in the Methods section (2.3.2) and in the Results section.

In the case of CFA, when considering possible structures, it would be important to illustrate and compare the fit of several solutions, e.g., unidimensional, correlated factors, second order, or bifactor models. The unidimensional model serves as a comparison. The bifactor structure includes the specific dimensions and the general factor, as used in the questionnaire evaluation (subscales and total score). We now see a correlated and second order models, although it would be more interesting to decide on the best model based on the fit of several models.

We tested a unidimensional model, which presented very poor overall goodness-of-fit indices (x2/gl=6.006; RMR=0.051; GFI=0.596; CFI=0.583; TLI=0.559; RMSEA=0.112), which is why a unidimensional solution was not chosen. Other factorial solutions were also sought, with four and six factors, but the most appropriate model was the five-factor one. Moreover, in the exploratory factorial analysis, the coefficients of Unidimensional Congruence, Explained Common Variance and Mean of Item Residual Absolute Loading suggested a correlated multifactor model.

Since the questionnaire's overall score is also used, I believe the second-order or bifactor models are much more suitable. The bifactor structure contains several specific factors (e.g. 5) and one general factor. See for example: Reise, 2012.

According to Marôco (2014), since the correlations between factors obtained in the correlational multifactor model are mostly high, this presupposes a second-order model.

The resulting fit indicators are poor, and it would be necessary to explain why correlations between errors exist. Modification indices often show that model fit would improve if one or more residuals among indicator variables were allowed to correlate. This practice is problematic for a variety of reasons. See for review: Hermida, R. (2015).

Thank you very much for this suggestion. We decided to present correlations between errors because, by carrying out a conceptual / theoretical analyses of some items, we concluded that some of them measured the same construct.

The fit index criteria you use are very lenient. More recently, though, much stricter indicators have been used. A CFI or GFI of 0.90 or below is not considered a good result. This will most likely cause problems when replicating the factor structure on another sample. „The results suggest that, for the ML method, a cutoff value close to .95 for TLI, BL89, CFI, RNI, and Gamma Hat; a cutoff value close to .90 for Mc; a cutoff value close to .08 for SRMR; and a cutoff value close to .06 for RMSEA are needed before we can conclude that there is a relatively good fit between the hypothesized model and the observed data. Furthermore, the 2-index presentation strategy is required to reject reasonable proportions of various types of true-population and misspecified models. Finally, using the proposed cutoff criteria, the ML-based TLI, Mc, and RMSEA tend to overreject true-population models at small sample size and thus are less preferable when sample size is small.”  Hu & Bentler, 1999.

In one of the more comprehensive and widely cited evaluations of cutoff criteria, the findings of simulation studies by Hu and Bentler (1999) suggest the following guidelines for acceptable model fit: (1) SRMR values close to .08 or below; (2) RMSEA values close to .06 or below; and (3) CFI and TLI values close to .95 or greater. Brown & Moore, 2012.

Thank you very much for your recommendation. We have included references to justify the fit index criteria we used in the Methods section. Moreover, we included some additional interpretations, based on those criteria, in the Results section.

Additionally, it would be beneficial to evaluate and demonstrate the normality of the variables, as this may also influence the method of factor analysis chosen.

In my opinion, the above-mentioned additional analysis would greatly increase the value of this study.

The normality of the variables was implicitly analysed as part of the data analysis procedures. We included a brief explanation about that in the Methods section.

References

Brown, T.A., Moore, M.T. (2012). Confirmatory factor analysis. Handbook of structural equation modeling. 361, 379

Hermida, R. (2015). The problem of allowing correlated errors in structural equation modeling: concerns and considerations. Computational Methods in Social Sciences, 3(1), 5-17.

Hu, L. T., & Bentler, P. M. (1999). Cutoff criteria for fit indexes in covariance structure analysis: Conventional criteria versus new alternatives. Structural equation modeling: a multidisciplinary journal, 6(1), 1-55.

Reise, S. P. (2012). The Rediscovery of Bifactor Measurement Models, Multivariate Behavioral Research, 47:5, 667-696. 

Tabachnick, B. G., & Fidell, L. S. (2007). Using multivariate statistics (5th ed.). Boston, MA: Allyn & Bacon.

Watkins, M. W. (2018). Exploratory factor analysis: A guide to best practice. Journal of Black Psychology, 44(3), 219-246.

Reviewer 2 Report (New Reviewer)

Introduction:

Some sentences need rephrasing. For example, Line 71: However, until the 1990s, this concept had not been put into practice. It is recommended to say something along these lines: However, the concept received limited attention until the 1990s.

Although the research questions are well-defined, please ensure the correct punctuation is used.

The primary concern of the introduction is the lack of a literature review and its relevance to its population, which has never been mentioned

Methods and study design

There are major flaws in this section.

The authors present the results in the methodology. (This psychometric study used a non-probability convenience with inclusion criteria, 122 such as participants aged 18 or older who freely agreed to participate. Thus, a sample of 123 401 participants aged between 18 and 88 years (M=43.0 years; SD=± 18.1) was recruited. 124 Of the participants, 58.1% were female, approximately 96.4% of the respondents were Por-125 tuguese, 97.6% were cohabiting, and more than half of the sample (53.9%) had completed 126 at least upper secondary education. In addition, 76.0% of participants were employed, 127 more than 61.0% had one child on average, and 68.8% lived in a city.)

However, the authors failed to present on how this study was conducted. For example, how were the participants recruited, and how bias was avoided. In addition, what were the inclusion and exclusion criteria for participating in this study?

Furthermore, what statistical analysis was conducted? The manuscript should have explained all these steps more concisely and presented very long statements for the Factor analysis rationale. The authors should consult manuscripts on how to write concisely for publications.

The authors stated that the team used SPSS statistical software for data analysis. However, they have 2 different versions of the software. Thus, which version they used?

The exploratory factor analysis was per-192 formed using version 25 of IBM SPSS, the principal component analysis and Varimax or thogonal rotation. Concerning factor retention decisions, eigenvalues greater than 1 and 194 the scree plot results were considered. IBM SPSS AMOS version 24 was used to conduct confirmatory factor analysis.

The manuscript stated the data was collected in 20828, 5 years ago. So why did the research team take such a long time to publish it?

The authors should have presented in what country this study.

Results: there are major flaws in this section.

Although the authors present the factor analysis in some detail, the results are not presented in a cohesively.

Discussion: There are major concerns about this section.

The authors do not state their results’ summary and do not present any correlation to other findings by interpreting and showing the value of this study. For example, This procedure is in line with Daniel [25](p. 136), arguing that "it might be advisable to delete items from a scale when it is being built or when one wishes to reduce the number of  items. Otherwise, the deletion procedure strictly inhibits comparing our results with the original scale."

What can the reader interpret from the above paragraph? What procedures are they presenting? What are Daniel’s findings that resonate with this study? Where was Daniel’s study conducted? Why is it relevant to this study?

Furthermore, some tables provide demographics and community level; however, they failed to provide any relevant information about the data.

The authors should have presented the limitations and strengths of this study.

Conclusion: There are significant failures. For example, the conclusion should be concise, emphasizing the need for additional study.

Although the authors mentioned that “These five dimensions can be applied to similar populations, by considering the raw

forms of the scales or the total scores obtained from the sum of the scores of the items  belonging to the different factors.”, they failed to state their studied population. Should the reader presume this study was conducted in Europe, Australia, or the USA?

There are many examples that need attention. For example, Line 71: However, until the 1990s, this concept had not been put into practice. It is recommended to say something along these lines: However, the concept received limited attention until the 1990s.

Author Response

Dear reviewer,

Thank you very much for your recommendations.

Introduction:

Some sentences need rephrasing. For example, Line 71: However, until the 1990s, this concept had not been put into practice. It is recommended to say something along these lines: However, the concept received limited attention until the 1990s.

Thank you very much for your suggestion. We have modified this sentence accordingly.

Although the research questions are well-defined, please ensure the correct punctuation is used. 

The primary concern of the introduction is the lack of a literature review and its relevance to its population, which has never been mentioned.

Methods and study design

There are major flaws in this section.

The authors present the results in the methodology. (This psychometric study used a non-probability convenience with inclusion criteria, 122 such as participants aged 18 or older who freely agreed to participate. Thus, a sample of 123 401 participants aged between 18 and 88 years (M=43.0 years; SD=± 18.1) was recruited. 124 Of the participants, 58.1% were female, approximately 96.4% of the respondents were Por-125 tuguese, 97.6% were cohabiting, and more than half of the sample (53.9%) had completed 126 at least upper secondary education. In addition, 76.0% of participants were employed, 127 more than 61.0% had one child on average, and 68.8% lived in a city.)

Thank you very much for having pointed out this mistake. We have moved this text to the Results section.

However, the authors failed to present on how this study was conducted. For example, how were the participants recruited, and how bias was avoided. In addition, what were the inclusion and exclusion criteria for participating in this study?

We have added further information on inclusion and exclusion criteria, as well as on bias avoidance. That information was included in the Methods section. We also added information regarding how and where data collection took place.

Furthermore, what statistical analysis was conducted? The manuscript should have explained all these steps more concisely and presented very long statements for the Factor analysis rationale. The authors should consult manuscripts on how to write concisely for publications.

All the data analysis procedures are presented, in detail, in section 2.3.2. We totally agree our data analysis options are not presented in a concise way. Nonetheless, the other reviewer asked us to include additional information regarding data analysis, that is why we cannot delete any information in this section. Nonetheless, we have tried to replace long sentences / statements by shorter ones.

The authors stated that the team used SPSS statistical software for data analysis. However, they have 2 different versions of the software. Thus, which version they used?

The exploratory factor analysis was per-192 formed using version 25 of IBM SPSS, the principal component analysis and Varimax or thogonal rotation. Concerning factor retention decisions, eigenvalues greater than 1 and 194 the scree plot results were considered. IBM SPSS AMOS version 24 was used to conduct confirmatory factor analysis.

In both cases we used version 25. We have already corrected that information. Thank you very much for your attention to these important details.

The manuscript stated the data was collected in 20828, 5 years ago. So why did the research team take such a long time to publish it?

During the COVID-19 pandemic the research considered it would be a priority to carry out research in that field, as we were facing a public health emergency. That is why we took such a long time to publish this paper.

The authors should have presented in what country this study.

We have added that information in the inclusion criteria as, indeed, living in Portugal was one the inclusion criteria we considered for this study.

Results: there are major flaws in this section.

Although the authors present the factor analysis in some detail, the results are not presented in a cohesively.

We have made some changes to this section in accordance with the recommendations presented by the other reviewer. Nonetheless, and once again, it was not possible to present this section in a more concise manner, as the other reviewer suggested us to add some additional results regarding factor analysis.

Discussion: There are major concerns about this section.

The authors do not state their results’ summary and do not present any correlation to other findings by interpreting and showing the value of this study. For example, This procedure is in line with Daniel [25](p. 136), arguing that "it might be advisable to delete items from a scale when it is being built or when one wishes to reduce the number of  items. Otherwise, the deletion procedure strictly inhibits comparing our results with the original scale."

What can the reader interpret from the above paragraph? What procedures are they presenting? What are Daniel’s findings that resonate with this study? Where was Daniel’s study conducted? Why is it relevant to this study?

We have added a brief summary of our results at the beginning of the Discussion section. We decided not to present all the psychometric properties of the instrument in there, as that would repeat the information that was already presented in the Results section. Regarding the paragraph we highlighted, we included additional information, so it may be easier for the reader to understand it.

Furthermore, some tables provide demographics and community level; however, they failed to provide any relevant information about the data.

Thank you for your suggestion. We decided to delete Table 7 as, indeed, it did not present any information that could be directly related to the development and/or validation of the instrument.

The authors should have presented the limitations and strengths of this study.

We included some strengths of the study at the beginning and at the end of the Discussion section. Moreover, we also included the limitations of the study at the end of the Discussion section.

Conclusion: There are significant failures. For example, the conclusion should be concise, emphasizing the need for additional study.

Although the authors mentioned that “These five dimensions can be applied to similar populations, by considering the raw 

forms of the scales or the total scores obtained from the sum of the scores of the items  belonging to the different factors.”, they failed to state their studied population. Should the reader presume this study was conducted in Europe, Australia, or the USA?

We have deleted some information in the Conclusion section, so it is now presented in a more concise way. The sentence you mentioned was deleted, as we considered it would not be relevant to present it in the Conclusion section. Nonetheless, we included information about the country in which the scale was developed and in which it may be used.

Round 2

Reviewer 1 Report (New Reviewer)

Thank you for your revision, and congratulations on your paper.

Author Response

Dear reviewer,

Thank you very much for having accepted our paper. Your recommendations were quite relevant to help improve its overall quality.

Reviewer 2 Report (New Reviewer)

Although the manuscript has been enhanced, the authors failed to present the survey and its analysis appropirately. There are significant concerns about grammar. For example, Line 60 through 74 is too long and convoluted.

Furthermore, there were no exclusion criteria. How was bias addressed?

Some of the graphs are problematic. For example, Table 1.  What re the authors presenting in Table 2?

There are areas where no references are cited. This is considered plagiarism. For example, line 90-92, no references.

Furthermore, what are the "uses Lluch Canut’s assumptions"?

There are significant concerns about grammar. For example, Line 60 through 74 is too long and convoluted.

What does "senior universities" mean

Author Response

Dear reviewer,

Thank you very much for your insightful comments / recommendations, which substantially help improve the overall quality of the paper.

Best regards.

Although the manuscript has been enhanced, the authors failed to present the survey and its analysis appropriately. There are significant concerns about grammar. For example, Line 60 through 74 is too long and convoluted.

Firstly, we would like to thank you very much for considering the manuscript has been enhanced. Regarding grammar, our paper was now entirely proofread by a professional translator. Thus, for instance, line 60 through 74 was improved according to your recommendation.

Furthermore, there were no exclusion criteria. How was bias addressed?

The aim of our study was to validate the Positive Mental Health Literacy Scale in a sample of adults in Portugal. Thus, we had to define some inclusion criteria such as, for instance, being 18 years or older, and living in Portugal. However, there was not any condition that could exclude a subject from being part of the sample. As we collected data by using paper-based questionnaires, we could avoid any kind of bias (e.g., the same participant answering the questionnaire twice). Moreover, even if the participant was not able to read the questions, the researcher could the questions and the potential / possible answers, so the participant could choose an answer. Thus, having a large sample and collecting data by using paper-based questionnaires were crucial strategies we used to avoid bias.

Some of the graphs are problematic. For example, Table 1.  What are the authors presenting in Table 2?

Regarding Table 1, indeed, the text presented before the table was not consistent with the results from the confirmatory factor analysis. Thus, we have corrected those inconsistencies. In Table 1 we presented the model fit indices, which resulted from the confirmatory factor analyses we carried out for each structural model. Those findings led us to conclude the second-order model was the best one.

Regarding Table 2, we have added footnotes that help understand the data we are presenting in there. It is important to note that, statistically, convergent validity is established when the Average Variance Extracted (AVE) is >0.50. That idea is explained before Table 2 (“As for AVE, the values observed do not indicate convergent validity as they show indices below 0.50, except for factor 3 - Perception of resources (AVE=0.546)”).

There are areas where no references are cited. This is considered plagiarism. For example, line 90-92, no references.

We included a reference in lines 90-92. Nonetheless, at that point we were not citing the author(s) but presenting a methodological option.

Furthermore, what are the "uses Lluch Canut’s assumptions"?

By “Lluch Canut’s assumptions” we meant “the six factors which comprise the Multifactor Model of Positive Mental Health”. Those factors had already been presented in the fourth paragraph of the “Introduction” section. Nonetheless, we added that information in the manuscript, in order to clarify that.

What does "senior universities" mean?

Indeed, that expression / term was not correctly translated into English. The entire paper was proofread by a professional translator, so that expression / term was replaced by “universities for senior citizens”.

This manuscript is a resubmission of an earlier submission. The following is a list of the peer review reports and author responses from that submission.

Round 1

Reviewer 1 Report

1.      The “Introduction” section should review the possible factors of the Positive Mental Health Literacy.

2.      On page 3, the review suggests deleting Table 1.

3.      Authors should provide the Positive Mental Health Literacy scale. Otherwise, the reviewer cannot check the items of the scale.

4.      On page 3, what is the direction of the scale’s items? How to obtain the 204 items is? What are the criteria about deleting the 70 items?

5.      The reviewer does not know the authors how to name the names of the factors. For example, factor 1 is named “initiative and decision -making”. The reviewer does not find any clue why the factor was named “initiative and decision -making”.

6.      Why the authors used the same samples for EFA and CFA?

Author Response

Dear reviewer,

We would like to thank you very much for your relevant and insightful comments / recommendations. We are sure they helped improved the overall quality of the paper, so we are grateful for that.

Reviewer 2 Report

This is appropriate and relevant topic and provides helpful insight about the thematic but specifically the lack of instruments available to assess Positive Mental Health Literacy levels in adult population, but some revisions are needed to be clearer and to improve the quality of the article.

Introduction

Overall, I feel that is necessary to go deep in some aspects and related better the main ideas and topics of the article. Also will be helpful to reformulate the introduction to be more fluid, coherent with the topic of the article and to become better related with discussion section. Going into details, and more specifically:

It is not clearly link the lines 34 to 48. Rephrase it, please;

Between lines 45 and 60 it is only refer one author. It is important to become broader your research and literature that corroborate what you are describing.

Between lines 62 and 70 the focus is in the concept of health literacy/mental health literacy and the need to develop assessment tools. This should appear first, because in previous paragraphs the authors already mentioned mental health literacy and different tools used in different setting and populations.

It is mentioned that the instrument focus on positive mental health literacy but this concept it is not explain or clarify through the introduction section. The concept of Positive Mental Health emerges from the concept of mental health and has its origins in Seligman and positive psychology and the work developed by Marie Jahoda. It is important to make this clearer and to add this references to the final list.

 It is not clearly and completed justify what you didn´t use the Positive Mental Health Questionnaire, also created by Lluch-Canut  but decided to create a new one.

 Methods

 Design, Participants and Setting  - it is not explain inclusion and/or exclusion criteria of the sample.

 Explain better how the cut-off of the groups was conducted and for the global score? This procedure allowed us to divide the participants into three 472 different levels according to their literacy scores: low, intermediate, and high.

This instrument can be used in younger populations? How to consider this?

Also authors should put the instrument in annex or in other document so the reviewers can check parts of the methods.

Was not conducted temporal stability (test-retest reliability) or concurrent validity, using instruments already designed and validated for the Portuguese population to measure positive mental health literacy. Why? This should be clear explained.

lines 117-118 – “The construction of the scale for assessing positive mental health literacy in adults 117 living in each community was based on the theoretical framework developed for this purpose and on the experience of researchers in scale construction”. Explain in more details what this means?

Discussion/conclusion: Overall, I feel that is necessary to go deep in some aspects related to the relation between the nuclear concepts. Also will be helpful to reformulate the introduction to relate better with discussion section. For example, lines 397-398: this is not clearly stated in introduction section. “Should be referred Its design was based on an interactive process that began with a comprehensive review of the national and international literature focusing on aspects related to positive mental health literacy”.

Organization of the paper, grammar, and references: Another read through to check grammar errors and format would be helpful. 

Author Response

(The authors gave the same response as above.)

Round 2

Reviewer 1 Report

At point 1, the information about the reply still does not enough.

The explanation of point 4 and point 5 still did not reply to my opinion.

About the authors' reply at point 6, the authors’ explanation did not touch the point because the purposes of the EFA and CFA are different.

In addition, the reviewer does not to understand the authors' reply about the meaning of no consensus on using two sub-samples.

Author Response

At point 1, the information about the reply still does not enough.

We have now included some additional information about the six factors proposed by the Lluch Canut’s positive mental health model, which was the main theoretical framework for this study.

The explanation of point 4 and point 5 still did not reply to my opinion.

We have better explained the procedure we carried out to produce 204 items and to delete 70 items afterwards. Moreover, we have tried to better explain how we named the factors of the scale. It is relevant to not the scale structure, which resulted from the EFA, was not in line with the six factors proposed by the Lluch Canut’s positive mental health model, and that is why we had to name the factors of the scale according to the semantic analysis of the items that composed each factor.

About the authors' reply at point 6, the authors’ explanation did not touch the point because the purposes of the EFA and CFA are different. In addition, the reviewer does not to understand the authors' reply about the meaning of no consensus on using two sub-samples.

Schmitt et al. (2018) pointed out that while it is often preferable to estimate the EFA and CFA model with different samples, especially for cross-validation, it is perfectly acceptable to fit different models to the same data in order to better understand the data generation process and factor structure.

Below you can find the reference which supports the abovementioned idea:

- Schmitt, T. A., Sass, D. A., Chappelle, W., & Thompson, W. (2018). Selecting the “best” factor structure and moving measurement validation forward: An illustration. Journal of Personality Assessment, 100(4), 345-362. https://doi.org/10.1080/00223891.2018.1449116